# Plasma Amino Acids and Acylcarnitines Are Associated with the Female but Not Male Adolescent Swimmer’s Performance: An Integration between Mass Spectrometry and Complex Network Approaches

**DOI:** 10.3390/biology11121734

**Published:** 2022-11-29

**Authors:** Flávio Marcio Macedo Mendes, Pedro Henrique Godoy Sanches, Álex Ap. Rosini Silva, Ivan Gustavo Masselli dos Reis, Patrícia de Oliveira Carvalho, Andréia M. Porcari, Leonardo Henrique Dalcheco Messias

**Affiliations:** 1Research Group on Technology Applied to Exercise Physiology—GTAFE, Health Sciences Postgraduate Program, São Francisco University, Bragança Paulista 12916-900, SP, Brazil; 2MS^4^Life Laboratory of Mass Spectrometry, Health Sciences Postgraduate Program, São Francisco University, Bragança Paulista 12916-900, SP, Brazil

**Keywords:** metabolomics, athletic performance, metabolism, critical velocity, swimming

## Abstract

**Simple Summary:**

Adolescent swimmers perform a lot of physical training with the aim of improving their performance in the sport. To achieve this goal, their diet must be adequate. In this scenario, studies have suggested that amino acids and acylcarnitines can improve the physical performance of athletes. However, we still do not have information as to whether the same occurs for adolescent swimmers. Thus, we compared amino acids and acylcarnitines present in the blood of adolescent swimmers of both sexes. We also correlated these substances with the performance of athletes. Our results showed that among amino acids and acylcarnitines, only tyrosine was lower in female adolescent swimmers when compared to male athletes. Additionally, significant associations between the female swimmers with tyrosine were observed. Future studies are necessary to understand this relationship, offering new possibilities for nutrition applied to the performance of adolescent swimmers.

**Abstract:**

The main aim of this study was to compare the performance over different distances, the critical velocity (CV), and plasma acylcarnitines/amino acids of male and female adolescent swimmers. Moreover, we applied the complex network approach to identify which molecules are associated with athletes’ performances. On the first day under a controlled environment, blood samples were collected after 12 h of overnight fasting. Performance trials (100, 200, 400, and 800-m) were randomly performed in the subsequent four days in a swimming pool, and CV was determined by linear distance versus time mathematical function. Metabolomic analyses were carried out on a triple quadrupole mass spectrometer performing electrospray ionization in the positive ionization mode. No difference was observed between the performance of male and female swimmers. Except for 200-m distance (*p* = 0.08), plasma tyrosine was positively and significantly associated with the female times during the trials (100-m, *p* = 0.04; 400-m, *p* = 0.04; 800-m, *p* = 0.02), and inversely associated with the CV (*p* = 0.02). The complex network approach showed that glycine (0.406), glutamine (0.400), arginine (0.335), free carnitine (0.355), tryptophan (0.289), and histidine (0.271) were the most influential nodes to reach tyrosine. These results revealed a thread that must be explored in further randomized/controlled designs, improving the knowledge surrounding nutrition and the performance of adolescent swimmers.

## 1. Introduction

Intense biological maturation and growth are characteristics of adolescence [1]. The physiological development during this period requires adequate macronutrient and micronutrient intake [2,3], which must be adjusted to meet the teenage athlete’s training and competition demands [4]. In an exercise-nutrition context, the science surrounding nutritional recommendations for the adolescent swimmer is present. While carbohydrates of high glycemic index of 1–1.2 g per kilogram of body mass per hour should be consumed after training, the protein supply during the same period must reach 1.2 g per kilogram of body mass per hour; regarding the daily fat intake, 2 g per kilogram of lean body mass are recommended [5]. However, competitive teenage swimmers can be exposed to massive physical training volumes [6], requiring adjustments in their diet to achieve better performance and growth development [7].

Beyond exercise demands, sexual dimorphisms for body composition, peak growth velocity, and pubertal development can cause male adolescents to consume more food than females [8]. In swimming, sex differences in body composition, energy, and kinematic variables of adolescent athletes have been described [9,10,11,12], but the specific nutritional patterns between male and female teenage swimmers and their influence on performance remain to be addressed. Studies within this population are limited to muscle protein metabolism [13], amino acid intake [14], and seasonal variations in training loads on selected amino acids [15]. Additionally, Hsueh [16] concluded that branched-chain amino acids (BCAA), arginine, and citrulline, improved adolescent swimming performances in a high-intensity interval protocol. Despite these advances, further cross-sectional studies are required to shed light on adolescent swimmers’ sex-specific nutritional patterns and their influence on performance.

In this context, acylcarnitines and amino acids have gained attention over recent years [17,18]. Carnitine content in the body is ~300 mg.kg^−1^ and is mostly distributed in the heart and skeletal muscle tissues, while only 0.5% is found in plasma [19]. Although its functions in intermediate metabolisms have been well documented, mainly regarding muscle fatty acid oxidation and glucose homeostasis [20], there is no scientific evidence to support that carnitine supplementation improves physical performance [17]. On the other hand, amino acids have been largely discussed in the ergogenic scenario [21,22,23,24]. Their role in muscle mass and health-related function is well established [25], although a recent meta-analysis observed minimal effects on the physical performance outcomes of healthy subjects [26].

To properly analyze the impact of physical performance on the acylcarnitines and amino acids plasma abundance, blinded, randomized, and controlled studies are necessary. However, without initial and cross-sectional reports, the wide range of possibilities can overshadow and delay scientific advances. The sportomics approach can advance the matter, since it has elucidated modulations generated by physical efforts in athletes [27,28,29,30]. Within a small biological matrix (e.g., blood, urine, tissue), thousands of molecules can be accurately identified and used to understand acute and chronic exercise effects [31,32,33]. Despite its application in cross-combat [34], soccer [35], and canoeing [36], the identification of metabolites by this method and its relationship with swimming performance is still at an embryonic stage.

Therefore, the main aim of this study was to compare swimming performances over four distances (100, 200, 400, and 800-m) and the critical velocity (CV) [37,38,39,40], using plasma acylcarnitines/amino acids from male and female adolescent athletes. Given the distinct pubertal development between male and female adolescents and their particular nutritional patterns, we hypothesized that significative differences in plasma acylcarnitines/amino acids would be observed between the athletes, which in turn could possibly be related to physical performance.

## 2. Materials and Methods

### 2.1. Subjects

Twenty male and eighteen female adolescent swimmers from regional levels were evaluated. The athletes had a daily training volume of 120 min a week. All swimmers had competed in regional and national competitions over the last two years. The Fédération Internationale de Natation points (i.e., FINA points) were calculated based on the period 01.09.2021–31.08.2022 and were obtained for males at 100-m = 214 ± 29, 200-m = 201 ± 21, 400-m = 205 ± 29, and 800-m = 202 ± 27, and for females at 100-m = 222 ± 16, 200-m = 212 ± 14, 400-m = 220 ± 15, and 800-m = 214 ± 16. Evaluations were conducted during the general preparatory period according to their training periodization. Informed consent was obtained from athletes and a parent and/or legal guardian. The experiments were approved by the Research Ethics Committee of the São Francisco University (24892219.3.0000.5514) and were conducted in agreement with the ethical recommendations of the Declaration of Helsinki.

### 2.2. Experimental Design

Swimmers were instructed to keep the same individual hydration/food habits throughout the experiment. No athlete reported the use of nutritional or ergogenic supplements. The swimmers completed five experimental sessions without performing any training sessions during the period. On the first day, under a controlled environment (laboratory facility), blood samples were collected after 12 h overnight fasting, and anthropometric measurements were performed in sequence. The performance trials were randomly performed in the subsequent four days in a swimming pool (25-m), 24 h apart. Efforts were initiated at the same hour (2:00 p.m.), and the exact order to dive off the block was maintained (Figure 1).

### 2.3. Metabolomic Analysis

The swimmers’ plasma was collected in a tube containing EDTA and stored in a freezer at −80 °C until analysis. The sample preparation followed the study of Sarafian [41]. All samples were thawed at room temperature and randomized before extraction to avoid analytical bias. A pooled sample was formed before sample extraction from equal parts of each sample (25 μL) and then aliquoted into different quality control (QC) samples, extracted with the swimmers’ samples. Plasma samples (150 μL) were randomized and cold isopropanol (200 μL) was added, followed by vortexing (30 seg) and centrifugation (12,000 rpm, 4 °C, 10 min). Then, the supernatant (200 μL) was collected and dried in N2. Blank samples were prepared using ultra-pure water instead of plasma. QC samples were distributed every 10 injections for instrumental monitoring, resulting in 10 QC samples for system suitability and 5 QC samples for intra-batch monitoring.

### 2.4. Tandem MS Analysis

Data acquisition was carried out on a Waters^®^ Xevo TQD triple quadrupole mass spectrometer (Waters Corporation, Milford, CT, USA) equipped with a Shimadzu^®^ SCL-10A controller, a Shimadzu^®^ LC-20AD pump controller, and a Shimadzu^®^ SIL-20A automatic sampler injector (Shimadzu Corporation, Kyoto, Japan). The procedures were adapted from Cicalini [42]. The analysis was performed using Flow Injection Analysis (FIA), with no chromatographic separation, using 10 µL as the injection volume. A flow gradient ranging from 0.01–0.50 mL/min was applied from 0.1 to 0.51 min and then kept for 3 min, after which the flow rate was decreased to 0.1 mL/min, resulting in 4 min of run time. The mobile phase was composed of water:acetonitrile:formic acid (80:20:0.1, *v/v/v*). The electrospray ionization source was operated in the positive ionization mode (ESI+) with the following parameters: desolvation gas flow of 850 L/h; source temperature of 150 °C; desolvation temperature of 500 °C; voltages of capillary and cone set at 3.0 kV and 60.0 V, respectively. The instrument was operated using the multiple reaction monitoring (MRM) mode, and the precursor > fragment transitions were optimized for each of the analyzed compounds (amino acids: n = 20; acylcarnitines: n = 12), as described in Appendix A. Peak areas of the transitions were recorded and integrated using Target Lynx software (Waters Corporation, Milford, CT, USA).

### 2.5. Performance Trials and Critical Velocity Assessment

The performance trials comprised distances of 100, 200, 400, and 800-m (4, 8, 16, and 32 turns, respectively). The warm-up was standardized in all tests and based on 10 min of stretching outside the pool followed by 5 min of swimming at low intensity. This warm-up is the same that the athletes perform before their training. Swimmers were instructed to provide their best performances during each effort. The time was recorded by the same researcher using a stopwatch and registered when the athlete touched the swimming pool edge. Commands such as “ready” and “go” were used and were standardized for all athletes. All tests were performed under a competition situation; one swimmer per lane. The CV was determined based on the four time trials. The linear equation D = CV t + AWC was applied for the assessment of the critical velocity (CV); where D is the distance covered (D), t refers to the time to cover the distance, CV relates to the slope of regression, and AWC relates to y-intercept. The CV is defined as the velocity that can be maintained without exhaustion [43,44,45], regularly associated with aerobic indexes [46,47,48,49]. The AWC represents a finite amount of work performed until exhaustion [47]. However, its physiological significance requires further investigation. Therefore, only CV and the 100, 200, 400, and 800-m times were considered for association with acylcarnitines and amino acid levels. The coefficient of determination (R²) was used to indicate the reliability of the linear adjustment.

### 2.6. Univariate and Multivariate Analysis

Statistical analyses were conducted using the STATISTICA 7.0 package (Statsoft, Tusla, OK, USA), MetaboAnalyst 5.0, and Python 3.9.3 environments. GraphPad Prism 5.0 software (GraphPad Software, San Diego, CA, USA) and the Canva environment were used to produce the figures. Pearson product-moment for the correlation analyses was performed in STATISTICA. Effect sizes (ES) were calculated at the following inferences: small if 0 ≤ |d| ≤ 0.5; medium if 0.5 < |d| ≤ 0.8; and large if |d| > 0.8).

The MetaboAnalyst environment [50] was used for most of the univariate and multivariate analyses. Data transformation (log transformation base 10) and scaling (Pareto) were adopted for the normalization of the data. We used the independent t-test followed by the false discovery rate (FDR) approach to compare the plasmatic levels of acylcarnitines/amino acids between male and female adolescent swimmers. Principal component analysis (PCA) and Partial Least-Squares Discriminant Analysis (PLS-DA) discriminated the male and female swimmers according to plasma acylcarnitines/amino acids. A random forest model was developed to predict the data into male or female swimmers based on these metabolite panels, and the out-of-bag score (OOB) was adopted to measure its prediction error.

A complex network topology analysis was created based on the significant (*p* < 0.05) correlations among the variables [51]. The variables (i.e., acylcarnitines/amino acids) were the nodes, and associations between these were represented by linking edges. As shown below in the results section, Tyrosine was the only molecule associated with the female adolescents’ swimmer performances. In this sense, a weighted and targeted complex network was created using Tyrosine as the target. In this analysis, both significant positive and inverse correlations were equally treated and received positive weights regardless of the correlation direction.

The edge weights were calculated as the product of the edge degree of proximity to the Tyrosine node (this can vary from 0.01 to 1; higher means closer) and the correlation coefficient between the nodes connected by the edge (this can vary from 0.01 to 1; higher is better). Therefore, edges received a weight equivalent to their respective correlation coefficient when they were directly linked to the node of interest (Tyrosine). Second-degree connections with the node of interest were equivalent to 0.5 (half) of the correlation coefficients, while third, fourth, and fifth-degree connections were equivalent to 0.250, 0.125, and 0.0625, respectively. Thus, the edge weights were used as the connection strength in the calculation of the target eigenvector scores. Centrality eigenvector values were obtained utilizing the NetworkX 2.5 library [52] inside Python. The eigenvector centrality for node i is the ith element of the vector x defined by the equation Ax = λx, where A is the adjacency matrix of the graph G with eigenvalue λ. There is a unique solution x, for which all entries are positive if λ is the largest eigenvalue of the adjacency matrix A [53]. The targeted eigenvector computed the centrality of a node based on the centrality of its neighbors and the weights of its edge connections.

## 3. Results

Male and female swimmers presented similar age (male = 15 ± 2 yr; female = 14 ± 2 yr; *p* = 0.348), body mass (male = 60.6 ± 11.4 kg; female = 54.9 ± 9.0 kg; *p* = 0.101), and height (male = 166 ± 16 cm; female = 160 ± 7 cm; *p* = 0.656). No difference was observed between swimmer performances over the predicted trials (100 m—male = 71.3 ± 10.4 s; female = 75.7 ± 5.4 s—ES = 0.53; 200 m—male = 166.3 ± 18.7 s; female = 174.1 ± 11.2 s—ES = 0.50; 400 m—male = 352.9 ± 53.9 s; female = 355.5 ± 25.1 s—ES = 0.06; 800 m—male = 743.9 ± 103.4 s; female = 752.4 ± 58.7 s—ES = 0.10) and the CV (male = 1.01 ± 0.10 m/min; female = 1.00 ± 0.14 m/min) (Figure 2). High regression coefficients were obtained for both groups (male—R² = 0.99 ± 0.00; female—R² = 0.99 ± 0.00).

Only tyrosine was significantly higher (1.56 ± 0.22 normalized intensity) in males when compared to females (1.30 ± 0.26 normalized intensity) among the amino acids (Figure 3A and Appendix A), and this difference was not able to discriminate the groups according to PCA or PLS-DA analyses (Figure 3B). Additionally, the OOB score was 0.289, with an error of 0.278 for females and 0.300 for males (Figure 3C).

A similar profile regarding the plasma acylcarnitines was observed between groups (Figure 4A and Appendix A). Moreover, these molecules were also not able to discriminate between male and female adolescent swimmers by PCA or PLS-DA analysis (Figure 4B). The classification error was 0.500 for females and 0.350 for males, and the OOB was obtained at 0.421 (Figure 4C).

Although no group differentiation was achieved, the significant difference observed for plasmatic tyrosine levels also revealed positive and significant correlations with the performance of 100, 400, and 800-m by female swimmers. Additionally, this amino acid was inversely and significantly associated with CV for the same group. By inspecting the correlation of tyrosine with other detected metabolites, glycine (0.406), glutamine (0.400), arginine (0.335), free carnitine (0.355), tryptophan (0.289), and histidine (0.271) were identified by the targeted network as being the most influential nodes to reach tyrosine (Figure 5). For the complete correlation matrix between the adolescent swimmer performances and the amino acids/acylcarnitines, see Appendix A.

## 4. Discussion

This is the first study to describe associations between female adolescent swimmer performances and plasma tyrosine. Furthermore, the eigenvector centrality showed that glycine, glutamine, arginine, free carnitine, tryptophan, and histidine may influence this molecule. Together, these results revealed a thread that must be explored in further randomized and controlled designs, improving the knowledge surrounding nutrition and the performance of adolescent swimmers.

Tyrosine (C9H11NO3) is a precursor to the catecholamines dopamine and norepinephrine. Its supplementation increases the circulating concentrations of these hormones in both the periphery and central nervous system [54]. When hormonal function is maintained intact, but dopamine and/or norepinephrine are temporarily depleted, tyrosine supplementation may enhance cognition [55]. Given the role of these hormones in motivation and thermoregulation [56,57,58,59,60,61], some research focus on the possible ergogenic activity of tyrosine on central fatigue and heat has been provided [62,63,64,65,66,67,68]. Furthermore, the role of tyrosine in body temperature is not limited to dopamine and norepinephrine synthesis. This amino acid is also a precursor of iodinated thyroid hormones [69], which exert crucial functions on many biological processes, including thermogenesis [70]. However, studies have drawn opposite conclusions regarding the tyrosine effect of physical exercise performed in heat [62,66,67,68]. Moreover, Sutton [71] showed that even with significantly increased plasma tyrosine levels by supplementation (150 g per kilogram of body mass consumed 30 min before performance tasks) the endurance, muscle strength, and anaerobic performance of moderately-highly active males were not improved.

Our results add to the omics-swimming area by demonstrating that the higher the tyrosine plasmatic levels of female adolescent swimmers, the worse the performance for 100 m, 200 m, 400 m, and the CV. Others have adopted the omics approach for swimmers, but with distinct purposes. By evaluating asthmatic and non-asthmatic adolescent swimmers using the breathomics approach, Couto [72] verified that metabolites associated with oxidative stress markers decreased after a swimming session, regardless of the group. Recently, Cai [73] identified serum metabolites (high-density lipoprotein, unsaturated fatty acid, lactic, methanol, isoleucine, 3-hydroxybutyric acid, acetoacetate, glutamine, glycine, and α-glucose) that discriminate between sub-elite and elite Chinese swimmers.

Based on the relationship between plasma tyrosine and female swimming performance, we conducted the targeted complex network approach to reveal connections between this amino acid and the remaining identified molecules. Plasmatic glycine, glutamine, arginine, and free carnitine were inversely correlated with tyrosine. Regarding this, a discussion exists regarding the latter three molecules on physical performance [17,74,75,76,77,78], including for swimmers [79,80,81]. While glutamine may promote glycogen synthesis and serve as fuel for lymphocytes and macrophages [73,82,83], carnitine is essential in beta-oxidation and can also modulate coenzyme-A metabolism [84,85]. Arginine is relevant for protein synthesis and can also be used to produce energy [86,87]. Histidine and tryptophan, on the other hand, were positively associated with tyrosine. Histidine may increase intracellular carnosine, offering an improved buffering capacity during exercise [88]. Being a precursor for serotonin synthesis, tryptophan supplementation has been deemed ergogenic for improving pain tolerance during exercise, but opposite results have also been drawn [89,90].

Arguments, coupled with the association of these molecules with tyrosine, suggest that some of these metabolites play an essential role in physical performance. However, this preliminary study could not detect the correlation between the plasmatic levels of glycine, glutamine, arginine, free carnitine, tryptophan, histidine, and the swimmer’s performance. On the other hand, since some of these metabolites participate in a similar biochemical process, it is possible to speculate on their indirect impact on female swimming performance. In this sense, the Pentose Phosphate pathway [91] leads to aromatic amino acids biosynthesis (tyrosine and tryptophan (r = 0.66)) and histidine hepatic catabolism [92,93]. In the liver, histidine takes its place in nitrogen metabolism, which catalyzes L-histidine in L-glutamate and NH3 [91], and is transported to the Urea Cycle via glutamine, to be finally converted to arginine, and participating in both nitrogen excretion and protein turnover [92].

Another point worth discussing is the difference in plasma tyrosine between male and female adolescent swimmers. Moller [94] verified that plasma tyrosine in healthy women was significantly reduced in the luteal phase when compared to the follicular. Recently, He [95] showed that plasma levels of alanine, glutamine, threonine, and tyrosine in eumenorrheic women significantly varied throughout the menstrual cycle, and significantly dropped to the lowest level close to Day 21. Although little data is available regarding amino acid fluctuations across the menstrual cycle, based on these studies, it is possible to suggest that the significant difference in plasma tyrosine of adolescent female swimmers can be associated with the menstrual cycle. However, given the age of athletes and the possible discomfort in providing this information for the research group, the phase or even the menarche attainment was not collected. Therefore, this explanation is merely speculative and requires further confirmation.

### Limitations and Future Perspectives

This study employed a descriptive and cross-sectional approach. Further studies are required to confirm that, when tyrosine levels are decreased, the physical performance of female adolescent swimmers improves. Moreover, blood samples were collected after 12 h overnight fasting before and on a distinct day of the performance trials. Future studies can advance this by collecting blood samples before and after swimming efforts to properly verify plasma tyrosine levels and their influence on a swimmer’s performance. Despite the important and new findings provided by the targeted network, it is not possible to assume a cause-and-effect relationship. Attention should also be paid to athlete levels. The tested swimmers were from regional to national levels; thus, our insights cannot be transposed to high-class swimmers. Another limitation was associated with the menstrual cycle and menarche attainment of female adolescent swimmers, as tyrosine levels were reported to change according to the menstrual cycle. Lastly, the critical velocity protocol is a non-invasive procedure; therefore, oxygen consumption and other classic parameters such as blood lactate were not collected. Further studies are warranted to associate the amino acids and acylcarnitines with these physiological data.

## 5. Conclusions

This study applied a sportomics approach and verified positive and significant correlations between tyrosine and female’s swimmer times for 100, 400, and 800 m, and inversely associated with CV. Moreover, the targeted complex network approach revealed that glycine, glutamine, arginine, free carnitine, tryptophan, and histidine may influence this molecule. However, these results, can only be transposed to female athletes. Future blinded, randomized, and controlled studies are necessary to advance on the new findings provided by this report, offering new avenues of investigation into the nutrition applied to the performance of swimmers.

## Figures and Tables

**Figure 1 biology-11-01734-f001:**
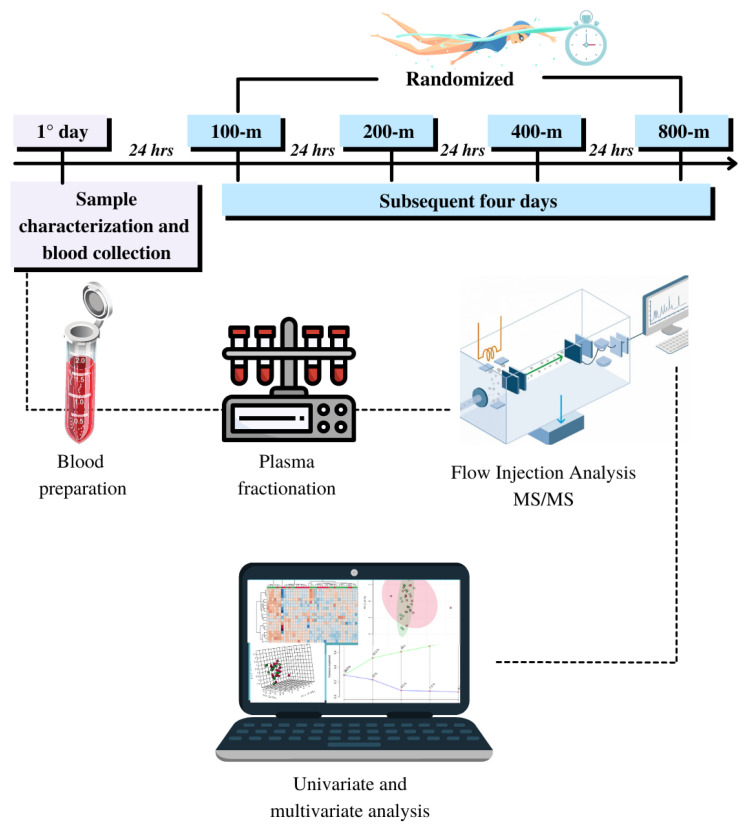
Experimental design of the study; MS—Mass Spectrometry.

**Figure 2 biology-11-01734-f002:**
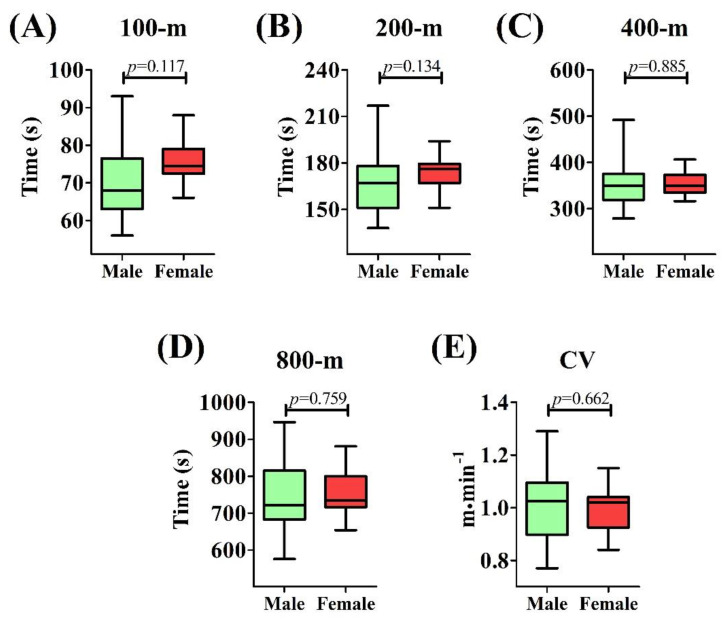
Comparison of performances over distinct distances and the critical velocity (CV) between male and female adolescent swimmers; (**A**) Comparison of performances on the 100-m; (**B**) Comparison of performances on the 200-m; (**C**) Comparison of performances on the 400-m; (**D**) Comparison of performances on the 800-m; (**E**) Comparison of CV; *p* ≤ 0.05.

**Figure 3 biology-11-01734-f003:**
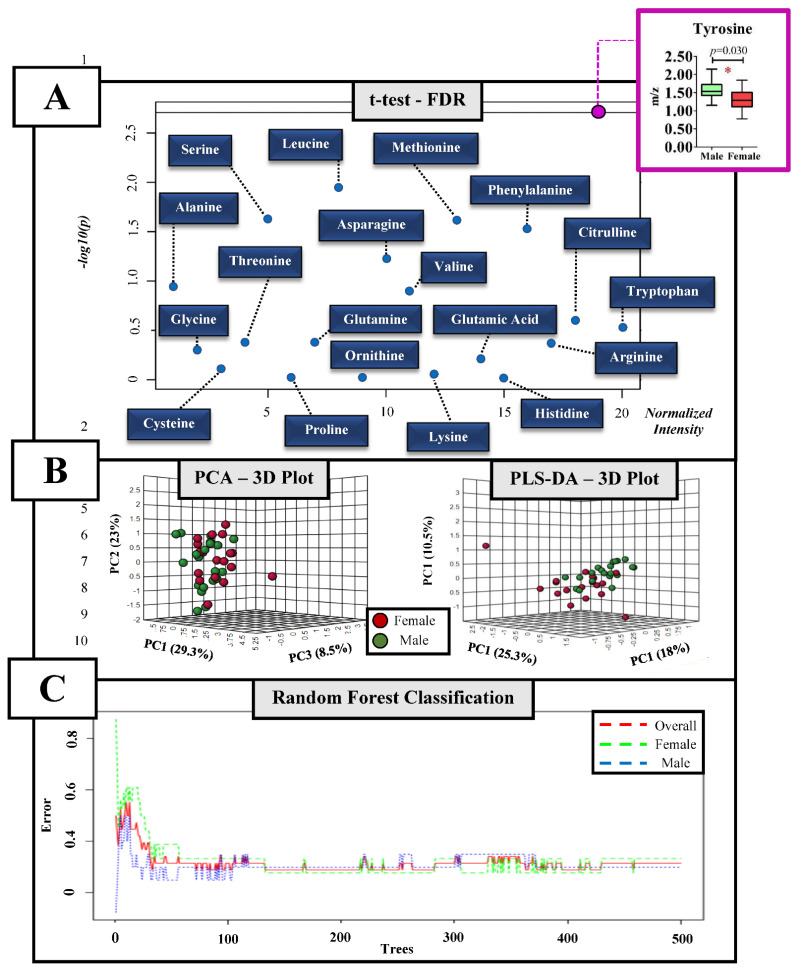
Univariate and multivariate analyses on the plasma amino acid of adolescent swimmers. (**A**) Independent *t*-test and False Discovery Rate (FDR) of plasma amino acids between male and female adolescent swimmers; (**B**) discriminatory approach by the Principal Component Analysis (PCA) and the Partial Least Squares-Discriminant Analysis (PLS-DA); (**C**) Random Forest classification; * Significant difference; *p* ≤ 0.05.

**Figure 4 biology-11-01734-f004:**
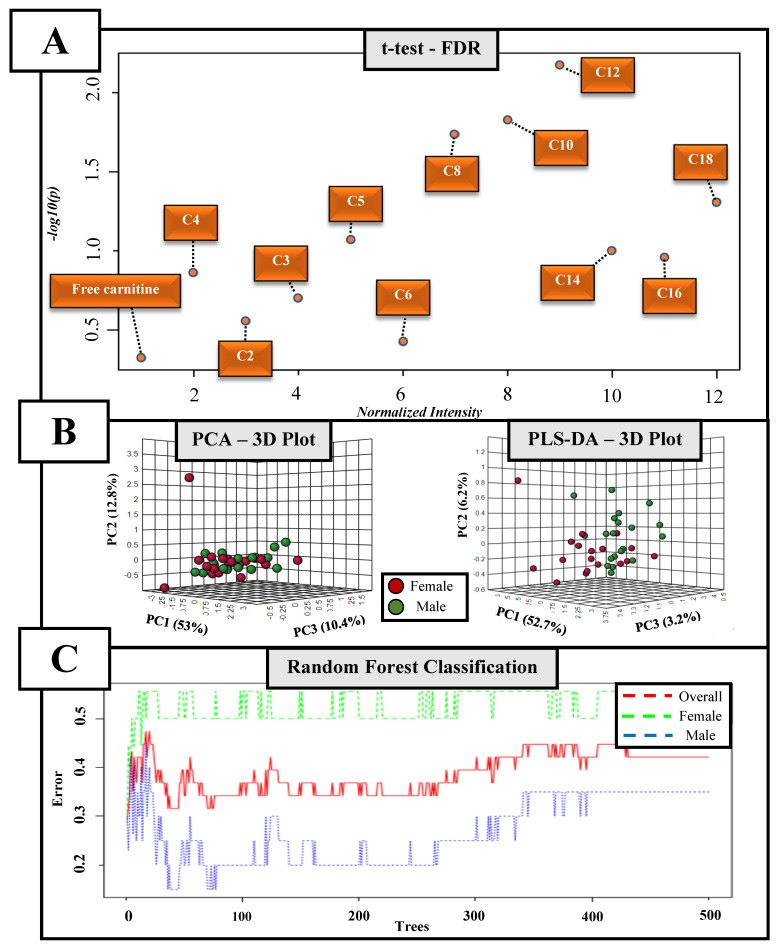
Univariate and multivariate analyses on the acylcarnitines of adolescent swimmers. (**A**) Independent t-test and False Discovery Rate (FDR) of plasma acylcarnitines between male and female adolescent swimmers; (**B**) discriminatory approach by the Principal Component Analysis (PCA) and the Partial Least Squares-Discriminant Analysis (PLS-DA); (**C**) Random Forest classification.

**Figure 5 biology-11-01734-f005:**
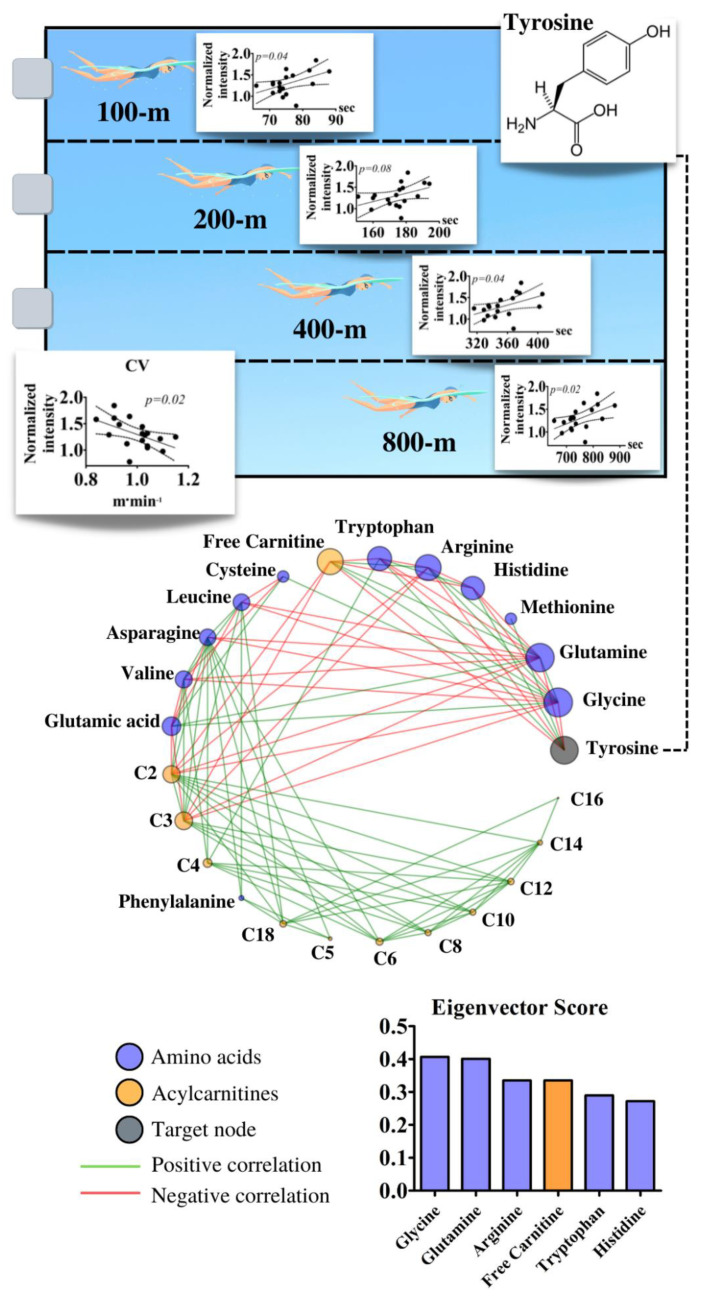
Upper panel presents the correlations between the performance trials for 100, 200, 400, and 800 m in addition to the critical velocity (CV) and plasma tyrosine of female adolescent swimmers. The lower panel presents the complex network based on the amino acids and acylcarnitines considering tyrosine as the target node.

## Data Availability

Not applicable.

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
