# Peer review of "Plasma Amino Acids and Acylcarnitines Are Associated with the Female but Not Male Adolescent Swimmer’s Performance: An Integration between Mass Spectrometry and Complex Network Approaches"

_biology, 2022, doi:10.3390/biology11121734_

Round 1
Reviewer 1 Report (Previous Reviewer 2)
At first, the reviewer has, of course, understood the experimental design of the study that the blood was collected only one point at 24 hours before the exercise experiments after 12-hour fasting. For most assessments for metabolic states, it is widely accepted, as the authors replied, that blood should be collected in fasting state for exception of the influence of diets. However, the authors must recognize that the exercise performance is strongly influenced by diets. In usually, athletes do not exercise in the fasting state after as long as 12 hours, do it? In the present study, the subjects took diets between the blood sampling and experiments as the authors replied. The authors replied that all subjects maintained the same nutritional habits during the evaluation days. However, the blood nutritional parameters immediately before the exercise experiment must be already difference to the values before 24 hours in the fasting state.
At second, the exercise such as done in the present study makes difference the nutritional parameters in blood. In the exercises after the 1st trial, the blood nutritional parameters must be influenced by the exercises carried out until the days before, in addition to the diets.
Based on the above concerns, the nutritional parameters collected before 24 hours at fasting state must not be related with the exercise performance that carry out in the subsequent four days. Therefore, the reviewer commented in the previous reviewing that it is nonsense.
So, the purpose is also unclear why the authors did try to evaluate the relationship between the nutritional parameters in fasting state before 24 hours and the subsequent repetitional exercise performance.
Reviewer suggests that the authors recognize these weak points, because the authors mentioned them in the limitation of the discussion section.
In the present study, there are many limitations that contain fatal; Only one blood sample was taken, and consequently, cause-and-effect relationship including blood lactate level could not be evaluated.
Finally, the fact that the conclusion states that further study is needed is itself an admission that the current study has not provided any useful information.
Because of study design, it is just a report of what was done, and there is nothing to be gained the new findings and insights as a result of the research.
Author Response
The answer has been entered in the file

Reviewer 2 Report (New Reviewer)
The article comprises an interesting matter, presents an according design and sample’s characterization. Despite a very simple design, were used proper protocols to collect and analyze data, showed through excellent images. The results were well described, and the discussion adequately interpreted it. Nevertheless, before being suitable for publication, there are some minor points to be analyzed by the authors, described below.
Despite the introduction brings a good scientific contextualization and clearly shows the rationale of the study, please consider a revision to organize the arguments and eventually make it short and more precise. Sometimes the arguments are unnecessarily repeated. Also, confirm the need of the text showed after lines 82. Despite not imperative, my suggestion is to remove it.
Please, explain at the article text the meaning of “bm-1 .h-1” that appears firstly at line 37.
Would be nice if the units of measurement were standardized. As an example, see line 55, that could be mg.kg-1 to align with saw at line 37. Also observe the line 391.
At line 382, please consider substitute “neurotransmitters” to “hormones” once your discussion is not anchored only on catecholamines as neurotransmitters.
Regarding tyrosine, it is also precursor of iodinated thyroid hormones, also associated to thermoregulation and eventually influencing physical performance. Please consider include some short discussion on it.
Author Response
The answer has been entered in the file

Reviewer 3 Report (New Reviewer)
Thank you for the opportunity to review this manuscript. This study represents a pilot investigation of the metabolomic characteristics of swimmers and their association with swimming performance. Despite such results being preliminary and needing further investigation to better understand their significance and potential application, it might be worth of publication to stimulate further research in this area providing early evidence and protocols.
Author Response
The answer has been entered in the file

Reviewer 4 Report (New Reviewer)
Introduction is well written. Miss the hypothesis of the study. Between L88 and L91, the information is related with results and should be deleted.
Methods: Clear.
Results: Clear and appelative with data visualization and easy understanding. The authors had a lot of work.
Discussion: Avoid "To the best of our knowledge".
References: Check numerations.
Author Response
The answer has been entered in the file

This manuscript is a resubmission of an earlier submission. The following is a list of the peer review reports and author responses from that submission.
Round 1
Reviewer 1 Report
Line 26: “…performances over the trials (100-m=0.04; 400-m=0.04; 800- 26 m=0.02) and the CV (p=0.02) only for female swimmers.” I guess these are p values. Be clear… Also, please inform 95%IC and effect sizes.
Line 27: “The complex network approach showed 27 that glycine (0.406), glutamine (0.400), arginine (0.335), free carnitine (0.355), tryptophan (0.289), and 28 histidine (0.271) as the most influential nodes to reach tyrosine” What are these values?
Line 41: “g/kg/b.w./h” …. Please, edit all metric units in your document. This example from line 41 is sounds very bad. Use “a·b-1” instead / … Change b.w. to BM (body mass).
Line 93: “Twenty male and eighteen female adolescent swimmers from regional levels were 93 evaluated. The athletes had a daily training volume of 120 minutes weekly.”
Weak… Insert more detailed information about your sample and training information...
Follow this study ( https://pubmed.ncbi.nlm.nih.gov/35894894/ ) while editing this section…
Also, I missed details about each swimmer’s pubertal maturation stage.
Line 101: “Experimental Design - Swimmers were instructed to keep the same individual hydration/food habits throughout the experiment. No athlete reported the use of nutritional or ergogenic supplements. The swimmers completed five experimental sessions. On the first day, under a controlled environment (laboratory facility), blood samples were collected after 12hrs overnight fasting, and anthropometric measurements were performed in sequence. The performance trials were randomly performed in the subsequent four days in a swimming pool (25-m), 24hrs apart. Efforts were initiated at the same hour (2:00 pm), and the exact order to dive off the block was maintained (Figure 1)."
I guess “24hrs apart” is not enough, or at least, a blood sample before 200, 400 and 800-m would be pertinent.
Line 144-150: “The time was recorded by a stopwatch and registered when the athlete touched the swimming pool edge. The linear equation D=CV*t + AWC was applied for the assessment of the critical velocity (CV), where D is the distance covered (D), t refers to the time to cover the distance, CV relates to the slope of regression, and AWC to y-intercept. The CV is defined as the velocity that can be maintained without exhaustion [42–44], regularly associated with aerobic indexes [45–48]. The AWC represents a finitet amount of work performed until exhaustion [46]. However, its physiological significance requires further investigation [49]."
I strongly suggest calculating the CV as proposed and tested in this study with similar population: https://pubmed.ncbi.nlm.nih.gov/26473520/ . Soon after, revise your results, discussion and conclusion.
Line 158: “Pearson product-moment for the correlation analyses was performed in STATIS- 158 TICA. “ Line 169 : “A complex network topology analysis was created based on the significant (p<0.05) 169 correlations among the variables [51].” and Supplementary Tables S2 and S3:
Most correlations were “negligible” or “weak” (doi: 10.1213/ANE.0000000000002864)
Reviewer 2 Report
The present study evaluated the relationship between several serum parameters and swimming performance in male and female adolescent athletes. Although the study carried out several statistical analyses, there are serious problems with the experimental protocol for evaluating the author's objectives.
Serum was collected only at 24 hours before the swimming exercise examination. Because several meals should be eaten between blood sampling and the first trial, plasma concentrations of amino acids and acylcarnitines should be significantly altered 24 hours later, just prior to the exercise test. In addition, 1-3 days were further passed during the exercise period, and consequently, the plasma parameters that are influenced by meals and exercises should be more altered the exercise period.
Therefore, it is nonsense to evaluate the relationship between the blood parameters examined at 24 hours before the exercise examination and the exercise performances during several days.
Reviewer 3 Report
This is a well-written study, with some interesting findings in adolescent swimmers, regarding tyrosine and performance. Congratulations on this interesting work. However, some specific comments are provided to clarify some issues.
- The main purpose should be more specific and clearer in the abstract. For instance, including the main variables to be assessed. Moreover, this should be also specified at the end of the introduction. The authors should be clearer according to the main aim. The aim is not only verified relationships but also to compare females and males, for instance.
Introduction:
- Very nice short introduction, as it should be, however, the relevance of the study should be better understood by the reader. Please, could you highlight the rationale of the study?
Material and methods:
- Please provide more characteristics of the sample training experience/level.
- Were they allowed to train during procedures?
- Please, provide more bibliographic references to support procedures used throughout the methods section.
- Only one researcher recorded the time during the trials. Official commands were used? Did they perform alone, or competition situation was provided? Please, provide more information on time trials. Some of this should be reported in limitations (discussion) if applied.
- What was the warm-up before time trials? Was always similar?
- The CV was determined based on two distances or the four-time trials? What were the distances used to determine CV?
Results
- Could you provide IC95% values for the difference between girls and boys? In some races, trials are so different in time (for example, 166 vs 174) but no differences were found.
- Very nice figures. Congrats. However, why did the authors use velocity as m.m-1 and no m/s as SI units?
Discussion
- Please include in the first sentence, “to the best of our knowledge”…
- Very nice discussion. Congrats.